# Prognostic Value of Lymphoid Infiltration and Aggregation in Cervical Cancer

**DOI:** 10.3390/cancers18010129

**Published:** 2025-12-30

**Authors:** Grace Gorecki, Macy Hale, Sarah Taylor, Geyon Garcia, Ian P. MacFawn, T. Rinda Soong, Tullia C. Bruno, Lan Coffman

**Affiliations:** 1Division of Hematology and Oncology, University of Pittsburgh, Pittsburgh, PA 15261, USA; grace.gorecki@ahn.org (G.G.);; 2Division of Gynecologic Oncology, Department of Obstetrics and Gynecology, and Reproductive Science, School of Medicine, University of Pittsburgh, Pittsburgh, PA 15213, USA; taylorse2@upmc.edu; 3Division of Hematology and Oncology, Department of Medicine, Hillman Cancer Center, University of Pittsburgh, Pittsburgh, PA 15232, USA; 4Department of Immunology, University of Pittsburgh School of Medicine, Pittsburgh, PA 15213, USA; 5Department of Biology, Grove City College, Grove City, PA 16127, USA; 6Department of Pathology, University of Pittsburgh Medical Center, Pittsburgh, PA 15213, USA; 7Magee Women’s Research Institute, University of Pittsburgh School of Medicine, Pittsburgh, PA 15213, USA

**Keywords:** cervical cancer, tertiary lymphoid structures, immunotherapy

## Abstract

The immune system plays an important role in cervical cancer, and immunotherapies are emerging as new treatment options in this disease. However, it is unclear which cervical cancer patients will respond to immunotherapy and how the makeup of immune cells within the cancer impacts the behavior of cervical cancer. In this work, we investigate the presence of specific immune structures, called tertiary lymphoid structures, within cervical cancer samples. We demonstrate that these structures are most beneficial when organized into well-defined clusters with specific signaling molecules such as CXCL13. Indeed, the presence of these immune structures with high CXCL13 levels is correlated with improved survival. This work highlights the importance of the spatial organization of immune cells within the cervical cancer environment and presents these findings as potential prognostic biomarkers and future targets to improve immunotherapy in cervical cancer.

## 1. Introduction

Cervical cancer is the most common gynecological cancer worldwide and is predominantly considered a virally mediated cancer, with more than 95% of cases being attributed to human papillomavirus (HPV) infection [1,2]. Cervical cancer can be diagnosed at an early stage with regular screening strategies such as Pap smears and high-risk HPV testing [3]. However, even when diagnosed at an early stage, 15–40% of patients have recurrence after primary treatment [4]. The advent of immunotherapy has revolutionized the treatment of multiple cancer types, including cervical cancer, where immunotherapy has shown promise, initially as a single-agent checkpoint inhibitor and more recently in combination with chemotherapy and chemoradiation for metastatic and locally advanced disease, respectively, where it demonstrates survival benefit [5,6,7]. However, disease response rates remain low, and survival in metastatic cervical cancer is unacceptably low, with a 5-year survival rate of less than 20%. Novel therapies are needed for those at risk for disease recurrence and/or metastasis. Indeed, critical gaps that need to be addressed in cervical cancer include improved methods to identify patients at risk for disease recurrence, biomarkers to predict response to immunotherapy, and improved targets to enhance the immune response.

Tumors with increased intratumoral cytotoxic T cells (CD8) and B cells have a strong association with improved survival in many cancers, including head and neck, breast, lung, and colon cancers and melanoma [8,9,10]. Chronic inflammation and infection can initiate an immune response in non-lymphoid tissues, forming ectopic tertiary lymphoid structures (TLSs). TLS are well-organized structures composed of immune cells clustered and organized into diverse structures with T cell zones, B cell follicles, and dendritic cells (DCs) that form in and around tumors. These structures can be categorized into three different states (lymphoid aggregates (LAs), TLS, and mature TLS), dependent on certain cellular and structural characteristics [11,12,13]. Early-stage TLS starts as LA, with disorganized clusters of B cells and T cells. Aggregates may form TLS comprising organized immune cells (T cells and B cells) and DCs, a process facilitated by the development of high endothelial venules (HEVs). The final state of TLS is distinguished by more complex features, such as follicular dendritic cells (FDCs) and germinal centers (GCs) bearing structural and functional resemblance to secondary lymphoid organs. TLS also plays a significant role in immune cell chemotaxis, which impacts cancer therapeutic response [9]. More specifically, CXCL13, a chemokine produced by stromal cells, Tfh cells, and CD8+ T cells, is known to localize in TLS and recruit immune cells, indicative of a role in TLS formation.

Recent work from our group demonstrated that HPV-positive head and neck squamous cell cancer (HNSCC) exhibited increased intratumoral TLS with germinal centers compared to HPV-negative HNSCC [9]. Given this finding and the strong relationship between HPV infection and cervical cancer, we sought to determine the prognostic significance of the immune microenvironment, particularly intratumoral B cells and TLS formation, in cervical cancer. Prior work indicates that most lymphoid structures in cervical cancer remain in an immature state without the formation of mature/active TLS. We hypothesized that lymphocyte infiltration and aggregation with the potential for TLS formation play a crucial role in the cervical cancer microenvironment and can serve as prognostic markers of survival in early-stage disease.

## 2. Methods

### 2.1. Patient Samples

We obtained formalin-fixed, paraffin-embedded (FFPE) tissue samples from 43 cervical cancer patients through the University of Pittsburgh’s ProMark tissue collection protocol and the Pitt Biospecimen Core under IRB (approval number: STUDY20050358, date of approval: 20 July 2020). All patients involved in this tissue collection undergo standard informed consent processes, and the study is continually reviewed by the institutional review board. All samples were deidentified by an honest broker system. All samples were reviewed and verified by a board-certified gynecologic pathologist.

### 2.2. TCGA Database and Survival Analysis

We used The Cervical Squamous Cell Carcinoma and Endocervical Adenocarcinoma (TCGA, Firehose Legacy) in a cohort of 310 patients. We analyzed the entire cohort, as well as isolated stage I cancer samples. Gene enrichment was defined as samples expressing mRNA 1.5 standard deviations greater than the rest of the cohort. The analyzed genes were CD19 or MS4A1 for B cells, CD8A for cytotoxic T cells, and CXCL13 for CXCL13. Then, 12-CK transcriptional signatures were measured for TCGA squamous cell cervical cancer samples by generating a score from the average expression of all genes in the signature. Patients were split by expression levels, and differences in survival were depicted with Kaplan–Meier survival curves and log-rank test *p*-values.

### 2.3. Immunohistochemistry (IHC)

The presence of B cells, CD8+ T cells, CXCL13, and P16 was analyzed using singleplex immunohistochemistry staining. Five-micron-thick FFPE tissue sections were deparaffinized, rehydrated, and subjected to antigen retrieval using heat-mediated antigen retrieval. Then, 3% hydrogen peroxide (Santa Cruz, sc-395440, Dallas, TX, USA) was used for blocking, and the sections were incubated with primary antibodies. After washing, the sections were incubated with biotinylated secondary antibodies, followed by streptavidin–peroxidase conjugate (Biolegend, San Diego, CA, USA). The sections were developed using 3,3′-diaminobenzidine (DAB) (Sigma, D8001, St. Louis, MO, USA).

### 2.4. Multispectral Staining and Imaging

Akoya Bioscience’s Vectra MOTIF (Akoya bioscience, Marlbobough, MA, USA) imaging pipeline and reagents were used for multispectral immunofluorescence experiments. Staining panels were optimized for Akoya’s MOTIF imaging platform, allowing for minimal spectral overlap and whole-slide imaging (Opals 480, 520, 570, 620, 690, and 780). Panels were optimized for autostaining on a Leica Bond Rx autostainer. Stained slides were scanned on a Vectra Polaris imager. Digital image files (QPTIFF) were processed with Phenochart software (Akoya bioscience, Marlbobough, MA, USA) to select ROIs for subsequent analysis. Whole-slide images were analyzed using QuPath (GNU General Public License v3.2).

### 2.5. Image Analysis

QuPath v3.2 image analysis software was used to quantify B cells, CD8+ T cells, and CXCL13 at the whole-slide level. Cell segmentation was used to determine the percentage of positive cells. The data generated by this analysis was paired with survival data. Patients were separated as ‘low infiltration’ or ‘high infiltration’ according to the median value. To determine the presence of lymphoid structures, a cell segmentation and a training pixel classifier were used to identify CD20-positive aggregates with a surface area over 50,000 microns^2^ and containing a minimum of 100 nuclei.

### 2.6. Statistical Analysis

Patient survival was analyzed using Kaplan–Meier and log-rank tests. Cox proportional hazards regression models were used to assess the independent prognostic value. All statistical analyses were performed using GraphPad Prism Version 10.2.3 (GraphPad Software, Boston, MA, USA) and STATA software (STATA 19, College Station, TX, USA) with a significance level of *p* < 0.05.

## 3. Results

### 3.1. CD8+ T Cell Infiltration Is Prognostic in Cervical Cancer. CD8+ T Cells and B Cells Colocalize in Regions of High Immune Cell Infiltration

We assembled a cohort of 43 patients diagnosed with clinical stage I treatment-naive cervical cancer (Table 1). After surgical resection, most patients remained staged as early cancer (31 patients), while 12 were upstaged to stages 2–4. Twenty-three patients had confirmed HPV+ disease on clinical pathology. We stained the remaining 19 patient samples for P16, a surrogate marker for HPV status, and confirmed all 19 were P16 positive (Appendix A). The median age was 42 years (26–70 years range), and 41% of patients had squamous histology, while 58% of patients had adenocarcinoma or mixed adenosquamous histology.

The infiltration of CD8+ T cells into tumors has positive prognostic value in many cancers, including esophageal, colorectal, breast, ovarian, head and neck, and cervical cancer [14,15,16,17,18,19,20]. Therefore, we first investigated the level of CD8+ T cell infiltration in our cohort using single-plex immunohistochemical analysis of FFPE tissue. We divided patients into two groups according to the median CD8+ T cell infiltration level as quantified using QuPath digital image analysis software (Figure 1a). Samples with CD8+ T cell infiltration below the median value were considered ‘low infiltrated,’ and samples with values above the median value were considered ‘high infiltrated’. Our analysis revealed that high infiltration of CD8+ T cells was prognostic for improved overall survival (OS). Median survival was 45 months for the low T cell infiltration group, while the high T cell infiltration group had median survival not reached (HR: 0.2766 [95% CI: 0.09956,0.7682]) (Figure 1b). When we analyzed the patients with surgically confirmed stage I disease, we found that the high infiltration group still had a trend towards improved outcomes, although the difference was not statistically significant due to the limited number of patients analyzed (*n* = 31) (HR: 0.3779 [95% CI: 0.1010,1.414]) (Figure 1c). The prognostic value of CD8+ T cell infiltration was particularly strong in the adenocarcinoma group, which is typically associated with worse outcomes. In those with adenocarcinomas, median survival was 58 months in the low CD8+ T cell infiltration group and was not reached in the high infiltration group (HR: 0.262 [95% CI: 0.05510,0.9283]) (Figure 1d). CD8+ T cell infiltration was not a statistically significant prognostic factor for recurrence-free survival (RFS) (Figure 1e–g).

We next looked at the infiltration of B cells in the cervical cancer microenvironment. Based on the median number of CD20+ cells, the cohort was divided into ‘high B cell’ vs. ‘low B cell’ groups. In contrast to our previous findings in HNSCC, we did not observe a significant association between B cell infiltration and survival outcomes in our cervical cancer cohort (Figure 1h). In patients with surgically confirmed stage I cervical cancer, we observed a trend towards improved overall survival in the high B cell infiltration group, although the difference was not statistically significant (HR: 0.5936 [95% CI: 0.1599,2.203]) (Figure 1i). Patients with adenocarcinoma histology also did not show significant survival improvement with high B cell infiltration, and median survival was not met for both ‘high B cell’ and ‘low B cell’ groups (Figure 1j). No association was observed in RFS (Figure 1k–m).

Next, we analyzed the spatial distribution of B cells and CD8+ T cells, and we observed that these cellular populations colocalized in regions where they were both highly infiltrated (CD20+ or CD8+ cells aggregated over 50,000 microns^2^ and 100 nuclei) (Figure 1n–s). There was an 89% colocalization rate of CD8+ T cell aggregates to B cell aggregates across all samples (Figure 1t).

### 3.2. Lymphoid Aggregates Are Positive Prognostic Factors When Associated with the Chemoattractant CXCL13+

To examine the impact of lymphoid aggregates in cervical cancer, we used spatial analysis to identify and quantify B-cell-rich structures in our samples. Tumor-infiltrating B cells (TIL-Bs) are associated with higher survival rates when organized into tertiary lymphoid structures (TLSs) [21,22]. Most of our samples contained at least one B cell aggregate (defined as CD20+ cells aggregated over 50,000 microns^2^ and 100 nuclei) (Figure 2a); however, no correlation was found between the quantity of structures and survival (Figure 2b). Given that lymphoid aggregates can vary widely in their organization and maturity, we next screened these structures for the expression of CXCL13 using IHC. CXCL13 is associated with B cell attraction and induces the formation of lymphoid aggregates [23]. We found that CXCL13 had the highest expression in regions of high CD8+ T and B cell infiltration (Figure 2d–g). We divided our cohort into ‘high’ and ‘low’ CXCL13 groups based on the median expression of CXCL13. High CXCL13 levels were associated with improved RFS, with a median survival of 53 months in the low infiltration group and not reached in the high infiltration group (HR: 0.3069 [95% CI: 0.1186, 0.7943]) (Figure 2i). There was also a trend toward improved OS in the CXCL13 high group, but this did not reach statistical significance (HR: 0.4358 [95% CI: 0.1580, 1.202]) (Figure 2h). The prognostic benefit for RFS was reproduced in the analysis of patients with surgically confirmed stage I disease (HR: 0.2436 [95% CI: 0.07267, 0.8165]) (Figure 2k). Patients with adenocarcinomas with high CXCL13 infiltration also showed a trend toward improved RFS (median survival of 52 months for the low infiltration group vs. not reached), although the results were not significant (HR: 0.3697 [95% CI: 0.1067, 1.281]) (Figure 2m).

Spatial analysis showed that expression of CXCL13 corresponded to specific LAs, suggesting these structures were indicative of more organized/mature TLS (Figure 2d–g). Therefore, we next analyzed samples that had a ‘high’ presence of LA (above median) with either ‘high’ or ‘low’ expression of CXCL13. Even though larger numbers of LA were not independently prognostic, the combination of high LA and high CXCL13 was significant for improved RFS: high LA and low CXCL13 demonstrated a median survival of 33 months, whereas the high LA and high CXCL13 group had a median survival that was not reached (HR: 0.2160 [95% CI: 0.04574, 1.020]) (Figure 2o). Our findings reinforce that it is paramount for B cells to be inserted into the right environment to favorably exercise a positive role in cancer outcomes.

### 3.3. High Infiltration with CD8 and CXCL13 Are Prognostic Factors in Cervical Cancer

To further determine factors associated with survival in our cervical cancer cohort, we conducted univariate and multivariable analyses. We included known clinical factors that are prognostic for disease recurrence based on SEDLIS criteria. SEDLIS criteria, which include depth of invasion, lymphovascular space invasion (LVSI), and tumor size, are used to determine the need for adjuvant radiotherapy in early-stage cervical cancer [24]. We also included age and histology as clinically important prognostic factors in addition to our above immune infiltration data (CD8+ T cells, CXCL13, and LA). In the univariate analyses, we found that high CD8+ T cell infiltration (HR: 0.288 [95% CI: 0.909, 0.091]) and increasing depth of invasion (HR: 1.096 [95% CI: 1.177, 1.020]) were independently prognostic for OS (Figure 3a), while high CXCL13 infiltration was independently prognostic for RFS (HR: 0.274 [95% CI: 0.853, 0.088]) (Figure 3b). In multivariable OS analysis, high CD8+ T cell infiltration and depth of invasion remained significant prognostic variables, and high CXCL13 infiltration became significant (HR: 0.079 [95% CI: 0.408, 0.015]) (Figure 3c). For the RFS multivariable analysis, CXCL13 remained significantly prognostic (HR: 0.181 [95% CI: 0.726, 0.045]), while histology (HR: 0.564 [95% CI: 0.971, 0.328]) and depth of invasion (HR: 1.129 [95% CI: 1.262, 1.010]) were also significantly correlated with RFS (Figure 3d). Hazard ratios for univariate OS, multivariate OS and univariate RFS are provided in Appendix A.

### 3.4. Multispectral Imaging Uncovers the TLS State Across the Cervical Cancer Cohort

Given that TLS activity is a spectrum and the more ‘active’ TLS marked by germinal centers (GCs) conveys the strongest survival benefit in other solid tumors, we wanted to understand the state and activity of TLS in our cervical cancer cohort. We used a multispectral immunofluorescence panel termed the ‘TLS activity’ panel, which we had previously defined to identify markers associated with the TLS state, and applied this to a subset of patient samples (*n* = 9) [12]. This panel consisted of CD4 (Biolegend, 300505, California, USA, CD20 (Biolegend, 302366, San Diego, CA, USA), CD21 (ThermoFisher, 17-0219-42, MA, USA), PNAd (Sigma, MABF2050, St. Louis, MO, USA), Ki67 (Biolegend, 350504, San Diego, CA, USA), and AID (Abcam, ab269454, Danvers, MA, USA) (Figure 4a). Single-plex images are shown in Appendix A. We followed the TLS activity scoring rubric previously developed by our group to classify lymphoid structures into three states (LA, TLS without GC, and TLS with GC) (Figure 4b). Of note, the presence of peripheral node addressin (PNAd) is a marker of high endothelial venules (HEVs), which play a crucial role in enabling the infiltration of blood-borne lymphocytes into the TLS; thus, it is a good indicator for the formation and immune activity of TLS [25,26]. TLS was defined as having a minimum of 50 CD20+ B cells, containing both CD20+ B cells and CD4+ T cells, and having PNAd+ HEVs. The presence of germinal centers (GCs) within TLS was distinguished by the presence of CD21+ FDC networks. We found that of our cohort, 56.04% were classified as lymphoid aggregates, 39.60% were classified as TLS, and only 4.36% were classified as TLS + GC (Figure 4c).

Furthermore, we phenotyped the immune population within the B cell aggregates in our cervical cancer cohort to elucidate the differences in TLS activity hallmarks across our groups. TLS + GC displayed a higher frequency of CD20+/Ki67+ cells, marking proliferating lymphocytes and CD20+/AID+ cells, indicative of affinity maturation of B cells (Figure 4d). Patients with increased TLS + GC and corresponding regions of affinity-matured B cells, however, did not show statistically significant improvement in OS and RFS (Figure 4e).

### 3.5. TLS and CD8+ T Cells Are Prognostic in the TCGA Cervical Cancer Validation Cohort

We next utilized publicly available mRNAseq data from TCGA to investigate the impact of TLS in a larger cohort of cervical cancer patients (*n* = 310). Most of the patients had squamous cell histology (*n* = 256), while 27 had adenocarcinoma, and 27 had other uncommon histologic subtypes, such as mucinous, adenosquamous, and endometrioid histology. In this cohort, 151 had stage I disease. To measure the prognostic benefit of TLS, we used the 12-chemokine (12-CK) transcriptional score (CCL2, CCL3, CCL4, CCL5, CCL8, CCL18, CCL19, CCL21, CXCL9, CXCL10, CXCL11, and CXCL13), which is a previously published predictive signature of TLS formation in tumors [27]. We analyzed the expression of each chemokine greater than 1.5 standard deviations above the mean and compared the OS, Stage I OS, and RFS of the group with a high 12-CK score to the unaltered group. The high 12-CK group demonstrated significantly improved OS in stage I patients compared to the unaltered group (Figure 5b, HR: 0.261 [95% CI: 0.120–0.567]). Similarly, in the whole cohort, there was a trend towards improved OS and RFS in the high 12-CK group, although these results were not significant (Figure 5a, OS HR: 0.451 [95% CI:0.227–0.894], RFS HR: 0.436 [95% CI: 0.203–0.935]). This data supports our findings that the presence of TLS is prognostic in cervical cancer.

We also analyzed survival in CXCL13, CD8+ T cell, and B-cell-enriched TCGA samples. Patients expressing CXCL13 greater than 1.5 standard deviations above the mean demonstrated a trend towards improved OS, Stage I OS, and RFS (Figure 5), but this did not reach statistical significance. Greater expression of CD8+ T cells had significant prognostic values for improved OS and Stage I OS (Figure 5). RFS was improved but not significant in patients expressing higher levels of CD8+ T cells (Figure 5i). Interestingly, the expression of B cell markers (MS4A1/CD19) showed a trend towards improved OS, Stage I OS, and RFS, but this also did not reach statistical significance (Figure 5j–l).

## 4. Discussion

Cervical cancer outcomes remain unacceptably low, and disease recurrence and metastasis are major drivers of morbidity and mortality. Immunotherapy is emerging as an important treatment in advanced-stage and recurrent disease, with anti-PD-1 immune checkpoint inhibitors [28]. However, even with these successes, the use of immunotherapy in earlier-stage disease is still being studied. Unlike in metastatic disease, recently published data investigating the utility of adding anti-PD-1 or anti-PD-L1 checkpoint inhibition to chemoradiation for locally advanced disease have not been shown to universally improve outcomes [7,29]. Understanding the complex interplay of the immune microenvironment in early-stage disease enables the possibility of refining risk stratification and improving outcomes. Our investigation underscores the role of lymphoid infiltration and the aggregation and organization into TLS as pivotal for tumor response in early-stage cervical cancer. Our results indicate that tumors demonstrating an immune-rich microenvironment, particularly those harboring active TLS, are associated with superior outcomes.

Tumor-infiltrating B cells have been correlated with improved outcomes in cancer [30,31]. In HSNCC, HPV+ tumors have a greater population of TIL-Bs than HPV- tumors [32,33,34], and HPV+ HSNCC is correlated with better outcomes [35,36,37]. Interestingly, total B cell infiltration was not prognostic in our cervical cancer cohort. This may be explained by the low level of B cells organized into ‘active’ TLS with GCs. Only 4% of the lymphoid aggregates were classified as TLC + GC.

Conversely, our analysis of CD8+ T cell infiltration emerged as prognostically significant in our cohort, a finding validated in a larger independent TCGA database. CD8+ T cell infiltration was also reported as prognostic for cervical cancer by other groups [20]. Previous research has demonstrated the critical role of CD8+ T cells in cell-mediated antitumor immune responses [38]. However, T cells can be evaded, deactivated, and suppressed within the tumor microenvironment, highlighting the importance of CD8+ T cell infiltration as a prognostic tool for the identification of patients who require immunotherapy.

Interestingly, our investigation revealed spatial colocalization of CD8+ T cells and aggregated B cells. This finding was a potential indicator that the regions where B cells and CD8+ T cells cluster were forming lymphoid aggregates or early-stage TLS [39]. The presence of both cell lineages has been associated with positive outcomes in ovarian cancer, melanoma, and sarcoma [22,40,41]. The organization of CD8+ T cells and B cells could be explained by the production of the chemokine CXCL13 by CD8+ T cells, which recruits CD20+ B cells to the site of CD8+ T cells [42]. Other studies suggest that activated intratumoral B cells chemoattract CD8+ T cells through the production of chemokines and are associated with the aggregation of both immune cells at high densities [32,43,44]. This suggests that CD20+ B cells and CD8+ T cells are bidirectionally recruiting one another to form lymphoid aggregates. Analysis of CXCL13 levels further suggested the presence of TLS in our samples. This chemokine was prognostic in colon cancer and related to TLS formation in ovarian cancer [41]. The association between CXCL13 and lymphoid aggregates was associated with a significant improvement in patient survival.

Collectively, these findings indicate that the mere detection of B cells in cervical cancer does not provide substantial insights into their prognostic significance. Instead, B cells organized in ‘active’ TLS and situated alongside other important immune cells (CD8+ T cells) with the appropriate chemoattractants (CXCL13) are necessary to impact survival outcomes.

To validate the observed outcomes as indicative of TLS, a multispectral image analysis was undertaken using our previously published ‘TLS activity’ panel [12]. The multispectral imaging uncovered that our cervical cancer cohort contained mostly lymphoid aggregates and lacked more active TLS with germinal centers. The lack of TLS + GC could possibly be due to an abundance of Tregs, regulatory T cells that suppress the immune response by inhibiting activation and proliferation of immune cells. Studies have found that in cervical cancer patients, there is an increase in Tregs over CD8+ T cells, while in head and neck cancer patients, there is a simultaneous increase in both CD8+ T cells and Tregs [45]. Notably, within these TLS + GC regions, the B cell population demonstrated enhanced proliferative and activation markers compared to lymphoid aggregates and TLS without GCs. Increased Tregs could inhibit the activation of B cells, preventing the formation of germinal centers and thus the affinity maturation of B cells. The patients who did contain TLS + GC and corresponding regions of affinity-matured B cells showed a trend towards improved OS and RFS. Similarly, MacFawn et al. [12] showed increased Tregs in ovarian versus fallopian tube lymphoid structures, corresponding with reduced B cell infiltration in ovarian lymphoid structures [12]. Therefore, immunotherapies to increase lymphoid infiltration, aggregation, and activity could be critical to improved survival in patients with cervical cancer. These may include therapies targeted at stromal and immune populations to improve recruitment, organization, and activation of TLS.

With the limited size of our cohort, we validated our results using data from a larger cervical cancer cohort from TCGA. In particular, 12-CK high scores are associated with the presence of TLS and are important in the inflammatory response, which can predict desirable outcomes [46]. Other studies have shown that high 12-CK scores can be positively related to improved response to cancer therapy [47,48,49]. Further validating the impact of TLS in cervical cancer, a high 12-CK score in the TCGA cervical patient cohort was prognostic for improved Stage I OS.

## 5. Conclusions

Collectively, this work underscores the pivotal role of the immune microenvironment in cervical cancer. Utilizing straightforward immunohistochemical (IHC) staining, we can identify individuals who may be at higher risk for disease recurrence or who may be optimal candidates for immunotherapy. Those with an immune-poor microenvironment face heightened risks of disease recurrence and poor treatment response. This highlights the considerable potential in enriching important immune populations and fostering the development of tertiary lymphoid structures (TLSs), which emerge as crucial immunotherapeutic targets.

Our data also suggests a potential role for utilizing immune-based prognostic features to identify patients who may benefit from early intervention with immunotherapeutics. However, it is important to acknowledge the limitations of our study, chiefly the statistical power constrained by the size of our patient cohort. Further investigations are needed to investigate the behavior of TLS in cervical cancer and to elucidate the effects of immunotherapy in this patient population.

## Figures and Tables

**Figure 1 cancers-18-00129-f001:**
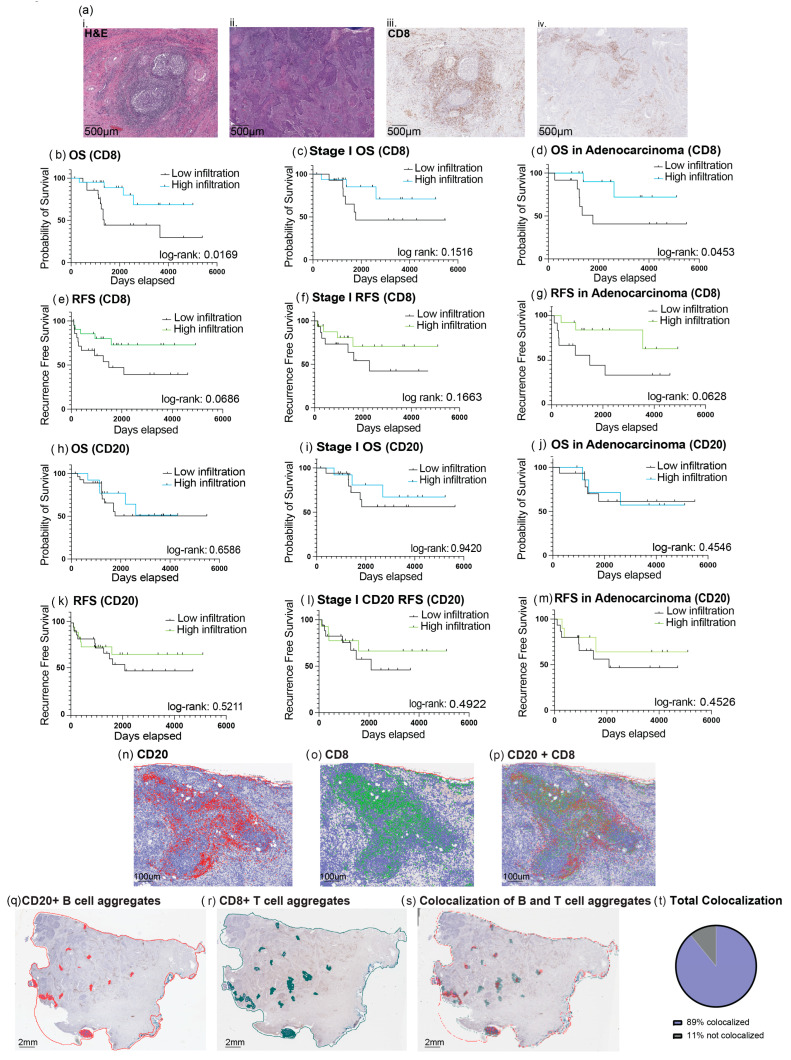
CD8+ T cell infiltration is prognostic in cervical cancer. CD8+ T cells and B cells colocalize in regions of high infiltration. (**a**(**i**,**ii**)) Representative images of H&E showing areas of tumor. (**a**(**iii**,**iv**)) CD8 immunohistochemistry staining performed on cervical cancer samples. (**b**–**g**) Kaplan–Meier plots comparing patients with ‘high’ and ‘low’ infiltration of CD8+ T cells for the whole cohort (**b**,**e**), Stage I patients (**c**,**f**), and adenocarcinoma patients only (**d**,**g**). (**h**–**m**) Kaplan–Meier plots comparing patients with ‘high’ and ‘low’ infiltration of CD20+ B cells for the whole cohort (**h**,**k**), Stage I patients (**i**,**l**), and adenocarcinoma patients only (**j**,**m**). (**n**–**p**) Representative images of cervical cancer samples with CD20 and CD8 immunohistochemistry, respectively, performed to highlight infiltration colocalization. Scale bars=100um (**q**) Areas of high infiltration of B cells. (**r**) Areas of high infiltration of CD8+ T cells. (**s**) Areas of high infiltration of B cells and high infiltration of CD8+ T cells overlaid to demonstrate colocalization. (**t**) Colocalization of CD8+ T cell aggregates to B cell aggregates across all slides (log-rank *p* value < 0.05 was considered significant).

**Figure 2 cancers-18-00129-f002:**
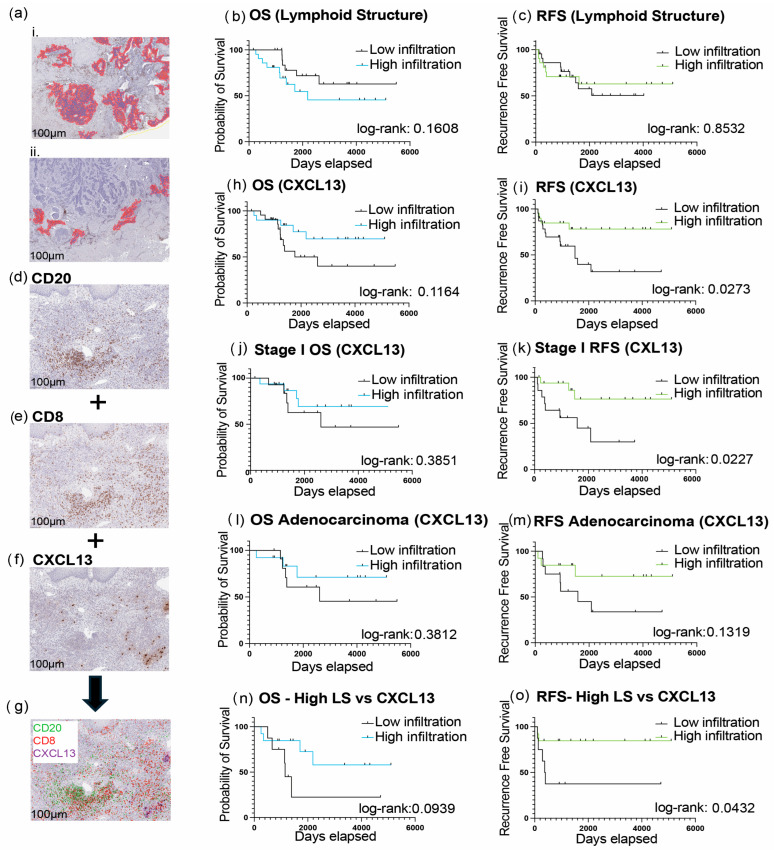
Lymphoid aggregates are positive prognostic factors when associated with a chemoattractant CXCL13+ environment. (**a**(**i**,**ii**)) Representative images of slide annotations in Qupath for regions containing CD20 lymphoid aggregates over 50,000 microns^2^ and 100 nuclei (purple), determined by trainable pixel thresholding. (**b**,**c**) Kaplan–Meier plots comparing patients with ‘high’ and ‘low’ occurrences of lymphoid aggregates. (**d**–**g**) Representative images of CD20, CD8, and CXCL13 immunohistochemistry staining performed on cervical cancer samples. (**h**–**m**) Kaplan–Meier plots comparing patients with ‘high’ and ‘low’ infiltration of CXCL13 for the whole cohort (**h**,**i**), Stage I patients (**j**,**k**), and adenocarcinoma patients only (**l**,**m**). (**n**–**o**) Survival curves of lymphoid aggregates paired with CXCL13 infiltration (log-rank *p* value < 0.05 was considered significant).

**Figure 3 cancers-18-00129-f003:**
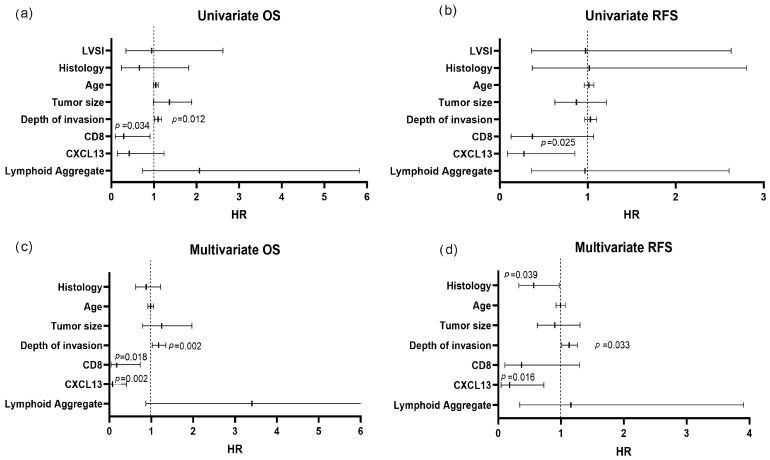
High infiltration with CD8 and CXCL13 is a prognostic factor in cervical cancer. (**a**,**b**) Univariate analysis of histology, age, tumor size, depth of invasion, LVSI, CD8, CXCL13, and lymphoid aggregate. (**c**,**d**) Multivariate analysis of histology, age, tumor size, depth of invasion, CD8, CXCL13, and lymphoid aggregate. (*p* < 0.05, log-rank test; Cox proportional hazard model).

**Figure 4 cancers-18-00129-f004:**
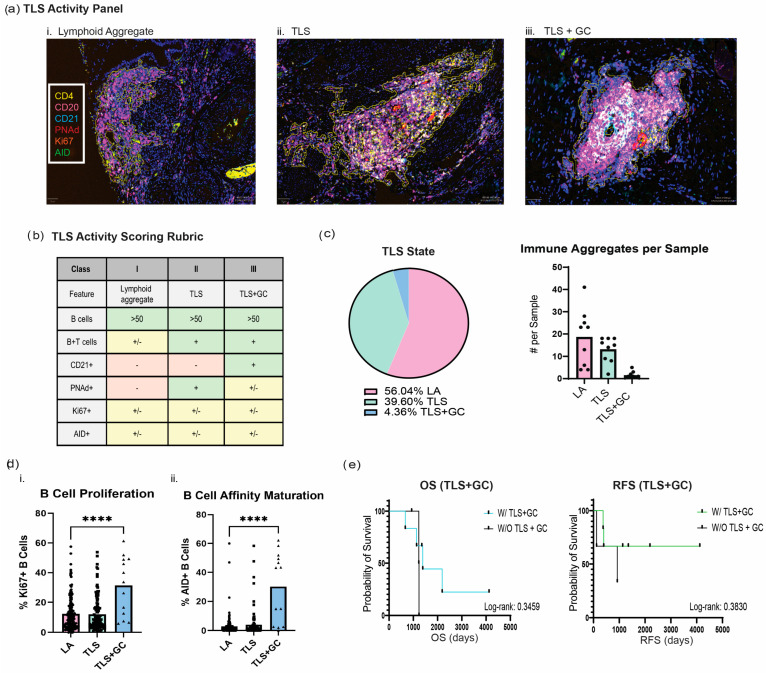
Multispectral imaging uncovers the TLS state across the cervical cancer cohort. (**a**) Multispectral images of the TLS maturity panel in a cervical cancer sample representative of (**i**) lymphoid aggregate, (**ii**) TLS, and (**iii**) TLS + GC. Scale bars=500um (**b**) TLS activity panel scoring TLS into three states (lymphoid aggregates, TLS, and TLS + GC). +/− refers to either presence or absence and aligns with this classification. (**c**) Total scoring of TLS state within the cervical cancer cohort (*n* = 9). (**d**) B cell proliferation and affinity maturation: (**i**) B cell proliferative Ki67+ abundance normalized to the total number of B cells. (**ii**) B cell affinity-matured AID+ abundance. **** = statistical significance (*p* < 0.05). (**e**) Kaplan–Meier plots comparing patients with TLS + GC and without TLS + GC.

**Figure 5 cancers-18-00129-f005:**
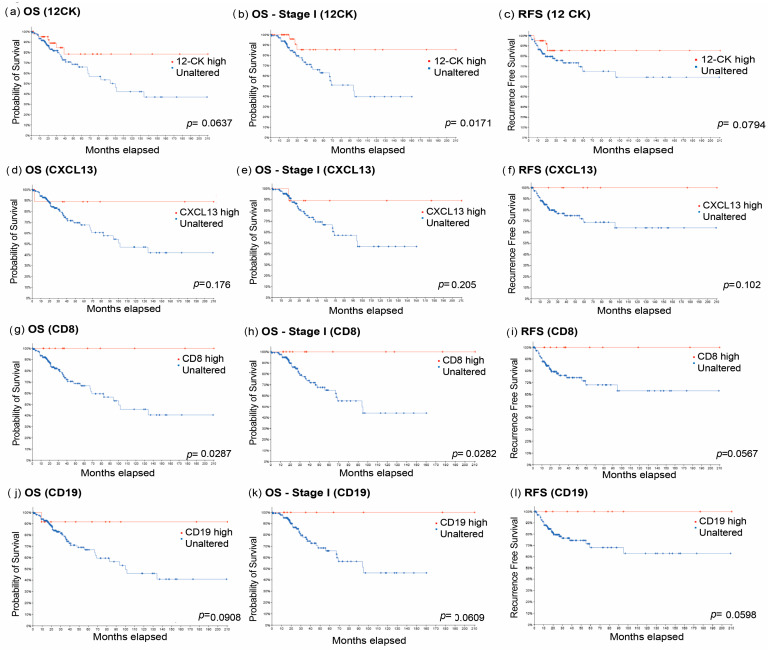
Our results were validated in a larger cohort (TCGA). TLS and CD8+ T cells remained prognostic in cervical cancer. (**a**–**l**) Kaplan–Meier plots comparing patients with ‘low’ and ‘high’ mRNA expression (threshold: >1.5 SD above the mean): (**a**) overall survival comparing expression of 12-CK score analyzing the whole cohort (high *n* = 51, unaltered *n* = 139), (**b**) only Stage I patients, high *n* = 42, unaltered *n* = 109, (**c**) recurrence-free survival (high *n* = 49, unaltered *n* = 123, (**d**) overall survival comparing expression of CXCL13 analyzing the whole cohort (high *n* = 11, unaltered *n* = 179), (**e**) only Stage I patients (high *n* = 10, unaltered *n* = 141), (**f**) recurrence-free survival (high *n* = 10, unaltered *n* = 162), (**g**) overall survival comparing expression of CD8A analyzing the whole cohort (high *n* = 12, unaltered *n* = 178), (**h**) only Stage I patients (high *n* = 13, unaltered *n* = 138), (**i**) recurrence-free survival (high *n* = 12, unaltered *n* = 160), (**j**) comparing expression of CD19/MSA41 analyzing the whole cohort (high *n* = 12, unaltered *n* = 178), (**k**) only Stage I patients (high *n* = 10, unaltered *n* = 141), (**l**) recurrence-free survival (high *n* = 11, unaltered *n* = 161) (log-rank *p* value < 0.05 was considered significant).

**Table 1 cancers-18-00129-t001:** Patient characteristics of the cohort, *n* = 43.

Patient Characteristics	Cohort, *N* = 43
Age at diagnosis (years)	
Median	43
Range	26–70
Histology	
Squamous cell carcinoma	18
Adenocarcinoma	25
FIGO stage	
I	31
II	5
III	6
IV	1
HPV status	
Positive	43
Negative	0
Recurrence-free survival (months)	
Median	44
Range	3–167
Number of recurrences	16
Overall survival (months)	
Median	47
Range	5–180
Number of deaths	15

## Data Availability

The data supporting the findings of this study are contained within the article and Appendix A. Data are available from the corresponding author upon reasonable request. TCGA data analyzed in this study are publicly available at https://www.cancer.gov/tcga (access date 4 August 2022).

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
