# Peer review of "Prognostic Value of Lymphoid Infiltration and Aggregation in Cervical Cancer"

_cancers, 2025, doi:10.3390/cancers18010129_

Round 1
Reviewer 1 Report
Comments and Suggestions for Authors
Hello.
Thank you for choosing me as a reviewer. I am particularly pleased with the work presented and well documented with images and conclusive statistical data. There are not many things to say because the graphics, the pathological anatomy images and the discussions, as well as the method used are eloquent for the proposed purpose and the results obtained
I am delighted to have reviewed such a work, which in my opinion achieves its proposed purpose and the conclusions are eloquent.
For young doctors, I consider it a welcome work for both gynecological oncologists and hematologists
Author Response
Reviewer: Thank you for choosing me as a reviewer. I am particularly pleased with the work presented and well documented with images and conclusive statistical data. There are not many things to say because the graphics, the pathological anatomy images and the discussions, as well as the method used are eloquent for the proposed purpose and the results obtained
I am delighted to have reviewed such a work, which in my opinion achieves its proposed purpose and the conclusions are eloquent.
For young doctors, I consider it a welcome work for both gynecological oncologists and hematologists
Response: Thank you for your thoughtful review of our work.
Reviewer 2 Report
Comments and Suggestions for Authors
The manuscript, “Prognostic Value of Lymphoid Infiltration and Aggregation in Cervical Cancer,” is potentially interesting, but the analyses could be more in-depth to yield novel results. I have the following comments:
- What is novel in this study? There are already published studies focused on the impact of TLSs on survival in cervical cancer patients (https://doi.org/10.1002/JLB.5MA0322-746R; doi: 10.18632/aging.205733; doi:10.1136/jitc-2025-012613). Similarly, CD8+ T cells were already analyzed in cervical cancer patients (see https://doi.org/10.1016/j.omto.2021.07.006).
- The number of patients included in the survival analyses is small. The results should be confirmed using a validation cohort of patients.
- Table 1 would be more informative if it also showed the number of patients who experienced relapse and those who died from the disease. These numbers are missing.
- CD21 is also expressed on B cells. How do you distinguish between B cell expression and FDC expression? Additionally, CD21 expression was also reported in TLSs with primary follicles; thus, it’s not a specific marker for GCs. Why did you not use AID?
- Why wasn’t the multispectral analysis performed in the whole cohort of patients, but only in 9 of them?
- In the Introduction, it is written that CXCL13 is expressed by CD8+ T cells; however, CXCL13 is mainly expressed by Tfh cells and stromal cells, like follicular dendritic cells. This should be corrected.
- The number of patients included in the study should be specified in Methods (chapter 2.1).
- In Chapter 2.4 in Methods, the multispectral staining should be described in more detail.
- It would be interesting to distinguish between the individual compartments and count the cells (like CD8+ T cells and CD20+ B cells) in the tumor epithelium and tumor stroma (and possibly in the invasive margin).
- In Fig. 1, the survival curves are shown for adenocarcinomas only (together with survival curves for all patients). Does that mean that adenocarcinomas were the drivers of the effect and that CD8+ T cells were not a prognostic marker in the case of squamous cell carcinomas?
- There are sentences in Results that belong to Discussion, like: “Co-localization of B and T cells is associated with more ‘active’ B cell organization and function [21-23]. CD8+ T cells have been shown to produce CXCL13, a chemokine that recruits immune cells, suggesting that CD8+ T cells may recruit B cells [24]. Therefore, the cell-to-cell interaction and colocalization pattern of CD20+ B cells and CD8+ T cells suggested that these cells are potentially bidirectionally recruiting one another to the same region and forming lymphoid aggregates.” Moreover, without the chemokine profile of these B and T cells, these suggestions are speculative.
- In Fig.4, no trend is visible, as stated in Results, lines 299-300 (usually a statistical difference with p < 0.1 is considered a trend).
- How many patients from the TCGA dataset were in the high group for each factor? Why median wasn’t used to separate the high vs. low samples in this case? It seems that only outliers were considered as high.
Author Response
Comments 1: What is novel in this study? There are already published studies focused on the impact of TLSs on survival in cervical cancer patients (https://doi.org/10.1002/JLB.5MA0322-746R; doi: 10.18632/aging.205733; doi:10.1136/jitc-2025-012613). Similarly, CD8+ T cells were already analyzed in cervical cancer patients (see https://doi.org/10.1016/j.omto.2021.07.006).
Response 1: The novelty of this work lies in the classification of TLS. Most prior publications focused on quantification of lymphoid aggregates/TLS without reference to activity/maturity state which is of critical importance. We are not only implementing multispectral imaging for accurate detection of TLS state based on published recommendations in the field (MacFawn et al. Cancer Cell 2024), but we are also defining the activity of TLS within each TLS state via Ki67 and AID. Overall, our manuscript not only interrogates TLS formation but also evaluates TLS state with activity. Similarly, we acknowledge CD8+ T cell infiltration has been previously reported in cervical cancer. We report it again in our cohort to confirm prior reports of prognostic importance and to complement our TLS analysis.
Comment 2: The number of patients included in the survival analyses is small. The results should be confirmed using a validation cohort of patients.
Response 2: We agree with the reviewer that our study has a limited sample size. We used TCGA expression data to validate markers associated with TLS. Future work will utilize multispectral analysis of TLS formation and activity in a larger group of patients but this is unfortunately not currently available.
Comment 3: Table 1 would be more informative if it also showed the number of patients who experienced relapse and those who died from the disease. These numbers are missing.
Response 3: We thank the reviewer for this comment. We have added these numbers into table 1 (15 deaths, 16 recurrences).
Comment 4: CD21 is also expressed on B cells. How do you distinguish between B cell expression and FDC expression? Additionally, CD21 expression was also reported in TLSs with primary follicles; thus, it’s not a specific marker for GCs. Why did you not use AID?
Response 4: We apologize for the lack of clarity. We exclude CD21 expression on B cells based on morphologic detection of FDC and CD20+ on B cells. We used machine learning to detect the network of FDCs. We did include staining for AID (figure 4) to mark GCs. While the expression is varied for FDCs and AID expression in GCs, we do demonstrate AID is significantly upregulated in TLS+GC (Fig 4d).
Comment 5: Why wasn’t the multispectral analysis performed in the whole cohort of patients, but only in 9 of them?
Response 5: We apologize for the lack of clarity. We chose the 9 samples that had increased lymphoid aggregates to perform Vectra multispectral analysis and further characterize the aggregates for TLS features. Without the presence of baseline lymphoid aggregates, further Vectra analysis would not be helpful.
Comment 6: In the Introduction, it is written that CXCL13 is expressed by CD8+ T cells; however, CXCL13 is mainly expressed by Tfh cells and stromal cells, like follicular dendritic cells. This should be corrected.
Response 6: We thank the reviewer for raising this issue. We acknowledge that CXCL13 is made by stromal progenitor and Tfh cells however, in humans CXCL13 is largely expressed by exhausted CD8+ T cells (as detailed in the following review: van der Leun et al. Nature reviews cancer. 2020). We have edited our introduction to include these details (page 2).
Comment 7: The number of patients included in the study should be specified in Methods (chapter 2.1).
Response 7: We apologize for this omission. The patient number (n=43) has been added to the methods section (page 3, section 2.1).
Comment 8: In Chapter 2.4 in Methods, the multispectral staining should be described in more detail.
Response 8: We have added further detail to the methods section in 2.4 page 3.
Comment 9: It would be interesting to distinguish between the individual compartments and count the cells (like CD8+ T cells and CD20+ B cells) in the tumor epithelium and tumor stroma (and possibly in the invasive margin).
Response 9: We agree with the reviewer that analysis by stromal vs tumor compartments would be interesting. Unfortunately, we did not perform this analysis in this cohort of patients. Future work will address the localization of individual immune cells as well as TLS in different tumor/TME compartments.
Comment 10: In Fig. 1, the survival curves are shown for adenocarcinomas only (together with survival curves for all patients). Does that mean that adenocarcinomas were the drivers of the effect and that CD8+ T cells were not a prognostic marker in the case of squamous cell carcinomas?
Response 10: We thank the reviewer for this question. Indeed, CD8+ T cell infiltration was prognostic for survival in the entire cohort and the adenocarcinoma cohort but not in the squamous cell cohort. This may be due to the limited number of patients as well as survival events (overall and recurrence free) given patients with squamous cell carcinoma generally have improved survival compared to adenocarcinoma patients. It is also possible that the adenocarcinoma cohort were the main drivers of the prognostic benefit of CD8+ T cells however histology was not a significant variable on either univariate overall survival or recurrence free survival nor was it significant in multivariate analysis for overall survival. Histology was a significant variable on multivariate recurrence free survival however (figure 3, page 9)
Comment 11: There are sentences in Results that belong to Discussion, like: “Co-localization of B and T cells is associated with more ‘active’ B cell organization and function [21-23]. CD8+ T cells have been shown to produce CXCL13, a chemokine that recruits immune cells, suggesting that CD8+ T cells may recruit B cells [24]. Therefore, the cell-to-cell interaction and colocalization pattern of CD20+ B cells and CD8+ T cells suggested that these cells are potentially bidirectionally recruiting one another to the same region and forming lymphoid aggregates.” Moreover, without the chemokine profile of these B and T cells, these suggestions are speculative.
Response 11: We thank the reviewer for pointing out this mistake. These sentences have been removed from the results section.
Comment 12: In Fig.4, no trend is visible, as stated in Results, lines 299-300 (usually a statistical difference with p < 0.1 is considered a trend).
Response 12: This has been corrected and changed to verbiage stating there is no statistically significant improvement in survival.
Comment 13: How many patients from the TCGA dataset were in the high group for each factor? Why median wasn’t used to separate the high vs. low samples in this case? It seems that only outliers were considered as high.
Response 13: We chose to use a threshold of 1.5 standard deviation above the mean to identify patients with the highest transcriptional levels of the evaluated parameters rather than simply looking at patients below and above the median. Simply evaluating based on the median value does not adequately separate patients indicating there is a higher threshold of immune activation necessary to impact prognosis in cervical cancer patients. This also emphasizes the transcriptional profile of the immune infiltrate is not as useful as spatial identification of immune cells within the TME. The number of patients in each group have been added to the figure 5 legend.
Reviewer 3 Report
Comments and Suggestions for Authors
Dear authors,
I congratulate you on your article, highlighting its exceptional quality and high methodological rigor. I have a few minor suggestions that, I believe, could further enhance the manuscript. -Page 1, Introduction, Paragraph 1: The sentence "The advent of immunotherapy has revolutionized treatment of multiple cancer types." (line 16) is somewhat generic. It could be improved with a brief example of immunotherapy's real impact in cervical cancer, even if summarized, to justify its "revolution" in this specific context. -Page 1, Introduction, Paragraph 2: The description of the three TLS states (LA, TLS, and mature TLS) is clear. However, it might be worthwhile to briefly reiterate that most structures in cervical cancer do not reach the most mature state, as will be detailed in the results. This could create a more precise expectation for the reader. -Page 5, Results, Paragraph 7: The paragraph's conclusion, "Our findings reinforce that it is paramount for B cells to be inserted into the right environment to favorably exercise a positive role in cancer outcomes." is a strong and well-supported statement. However, a brief insight into how this could be therapeutically achieved (e.g., with chemokines like CXCL13) might enrich the perspective.Author Response
I congratulate you on your article, highlighting its exceptional quality and high methodological rigor. I have a few minor suggestions that, I believe, could further enhance the manuscript. -
Comment 1: Page 1, Introduction, Paragraph 1: The sentence "The advent of immunotherapy has revolutionized treatment of multiple cancer types." (line 16) is somewhat generic. It could be improved with a brief example of immunotherapy's real impact in cervical cancer, even if summarized, to justify its "revolution" in this specific context.
Response 1: We thank the reviewer for this suggestion. We have edited this sentence as follows "The advent of immunotherapy has revolutionized treatment of multiple cancer types including cervical cancer where immunotherapy has shown promise, initially as a single agent check point inhibitor and more recently in combination with chemotherapy and chemoradiation for metastatic and locally advanced disease, respectively where it demonstrates survival benefit"
Comment 2:-Page 1, Introduction, Paragraph 2: The description of the three TLS states (LA, TLS, and mature TLS) is clear. However, it might be worthwhile to briefly reiterate that most structures in cervical cancer do not reach the most mature state, as will be detailed in the results. This could create a more precise expectation for the reader.
Response 2: We appreciate this comment and have added the following sentence "Prior work indicates most lymphoid structures in cervical cancer remain in an immature state without formation of mature/active TLS."
Comment 3: -Page 5, Results, Paragraph 7: The paragraph's conclusion, "Our findings reinforce that it is paramount for B cells to be inserted into the right environment to favorably exercise a positive role in cancer outcomes." is a strong and well-supported statement. However, a brief insight into how this could be therapeutically achieved (e.g., with chemokines like CXCL13) might enrich the perspective.
Response 3: We thank the reviewer for this suggestion. We have added the below sentences into the discussion to address this. "Therefore, immunotherapies to increase lymphoid infiltration, aggregation and activity could be critical to improved survival in patients with cervical cancer. These may include therapies targeted at stromal and immune populations to improve recruitment, organization and activation of TLS."
Reviewer 4 Report
Comments and Suggestions for Authors
Another proof that the immune system is crucial in the fight against cancer.
Cervical carcinoma specimens that are rich in immune cells have had better prognosis and less chance for recurrence/metastasis. Survival curves also matched with more immune cell infiltration.
Although the observations are very logic and coincide with the literature, The value of this study would be further magnified if a clinical trial based upon these findings could be planned.
Author Response
Comment 1: Although the observations are very logic and coincide with the literature, The value of this study would be further magnified if a clinical trial based upon these findings could be planned.
Response 1: We agree with the reviewer and future work is being planned to incorporate TLS assessment into clinical trials with immunotherapy in cervical cancer as a potential predictive biomarker. Additionally, we are actively working on potential therapeutic targets to enhance TLS in multiple gynecologic cancers.